Prey aggregation is an effective olfactory predator avoidance strategy

Johannesen Asa 1 2 asajoh@fiskaaling.fo
Dunn Alison M. 2
Morrell Lesley J. 3
1 Marine Centre, Fiskaaling , Hvalvík , Faroe Islands
2 School of Biology, University of Leeds , United Kingdom
3 School of Biological, Biomedical and Environmental Sciences, University of Hull , United Kingdom
Hay Mark
Electronic publication date: 2014 May 27
Publication date: 2014
Volume: 2
Electronic Location ID: e408
Received 2014 Mar 27; Accepted 2014 May 10
Copyright: © 2014 Johannesen et al.
Copyright year: 2014
Copyright holder: Johannesen et al.
License: This is an open access article distributed under the terms of the Creative Commons Attribution License, which permits unrestricted use, distribution, reproduction and adaptation in any medium and for any purpose provided that it is properly attributed. For attribution, the original author(s), title, publication source (PeerJ) and either DOI or URL of the article must be cited.
License URL: https://creativecommons.org/licenses/by/4.0/

Keywords: Olfaction, Aggregation, Predator–prey interactions, Stickleback, Gasterosteus aculeatus, Prey detection

Funding: Faroese Research Council This work was funded by the Faroese Research Council as part of a PhD stipend for Asa Johannesen. The funders had no role in study design, data collection and analysis, decision to publish, or preparation of the manuscript.

==============================
Predator–prey interactions have a major effect on species abundance and diversity, and aggregation is a well-known anti-predator behaviour. For immobile prey, the effectiveness of aggregation depends on two conditions: (a) the inability of the predator to consume all prey in a group and (b) detection of a single large group not being proportionally easier than that of several small groups. How prey aggregation influences predation rates when visual cues are restricted, such as in turbid water, has not been thoroughly investigated. We carried out foraging (predation) experiments using a fish predator and (dead) chironomid larvae as prey in both laboratory and field settings. In the laboratory, a reduction in visual cue availability (in turbid water) led to a delay in the location of aggregated prey compared to when visual cues were available. Aggregated prey suffered high mortality once discovered, leading to better survival of dispersed prey in the longer term. We attribute this to the inability of the dead prey to take evasive action. In the field (where prey were placed in feeding stations that allowed transmission of olfactory but not visual cues), aggregated (large groups) and semi-dispersed prey survived for longer than dispersed prey—including long term survival. Together, our results indicate that similar to systems where predators hunt using vision, aggregation is an effective anti-predator behaviour for prey avoiding olfactory predators.

Introduction

Predator–prey interactions are one of the major factors influencing patterns of species diversity and abundance in ecosystems (Chesson & Kuang, 2008). Predators influence prey abundance and distribution through both consumptive and non-consumptive effects (Preisser, Orrock & Schmitz, 2007) such as predator avoidance behaviours, which may limit prey access to resources (Griffiths & Richardson, 2006). Aggregation into groups is a common response to the risk of predation (Krause & Ruxton, 2002). Grouping individuals benefit from the dilution effect if a predator is unable to consume all prey in a group (Foster & Treherne, 1981) and from encounter dilution, where aggregated prey are encountered less often assuming population size is kept constant (Wrona & Dixon, 1991). Together, this leads to a situation where fewer predators survive because cost of finding a prey group is high, and more prey survive because predators only consume a few prey per encounter (Turner & Pitcher, 1986; Turesson & Brönmark, 2007).

Prey detection is likely to be dependent on a predator’s sensory acuity and modality (Cain, 1985). As a group of prey grows, the ability of a visual predator to detect the group is predicted to increase at a slower rate; that is, a group of N individuals should be less than N times more detectable than a single individual (Brock & Riffenburgh, 1960; Treisman, 1975; Turner & Pitcher, 1986). This is supported by empirical evidence for visual predators, where a non-proportional relationship between prey group size and detection rate has been found for humans seeking computer generated prey (Jackson et al., 2005), sticklebacks (Gasterosteus aculeatus) attacking Daphnia swarms (Ioannou et al., 2011) and great tits (Parus major) searching for aposematic prey (Riipi et al., 2001).

Whether encounter-dilution effects operate when predators use other sensory modalities is unclear. Close neighbours are likely to produce odour plumes that interact, increasing both the area of the odour plume and the amount of stimulant (Monismith et al., 1990). Treisman (1975) suggests that a group of N individuals should be detectable by an olfactory predator at a distance N times as great as that for a single prey, resulting in an area in which the group can be detected N2 times as large as for a single prey (or a volume N3 times as large). If this is the case, encounter-dilution would not take place, and grouping would not be favoured unless the predator is highly sensitive to olfactory cues and does not preferentially target large groups over small ones (Cain, 1985). Recent empirical data indicates that aggregation increases risk of predation by olfactory predators (Whitton et al., 2012; Wilson & Weissburg, 2012) but Andersson et al. find that the distance at which a group can be detected increases asymptotically with group size (Andersson, Löfstedt & Hambäck, 2013).

While patterns of risk with increasing levels of aggregation are beginning to be established, there is no work that directly contrasts visual and olfactory prey detection rates on dispersed and aggregated prey within the same predator. Changes in the environment, such as fluxes in turbidity or changes in pH, can alter the availability of visual and olfactory information (Leduc et al., 2013). Consequently, these can alter reliance on different sensory modalities by predators (Chapman et al., 2010), which in turn may affect the shape of the interaction between predators and prey. Predators may use both vision and olfaction in detecting prey, increasing reliance on olfaction under poor visual conditions (Chapman et al., 2010). We predicted that the benefits of aggregation as an anti-predator defence would be reduced or eliminated when predators hunt using olfaction rather than vision. To test this prediction, we investigated the ability of sticklebacks (Gasterosteus aculeatus) to detect and consume dispersed and aggregated bloodworm when visual cues were and were not available. Chironomid larvae coexist with sticklebacks in the wild and are often used as a commercial fish food. The diet of the captive sticklebacks used in this study consisted solely of bloodworm. Sticklebacks are often found in waters that are highly variable in turbidity (Wootton, 1976) and employ olfaction to detect prey in turbid water to compensate for the loss of visual cues (Johannesen, Dunn & Morrell, 2012). As a measure of detection, we monitored the survival of prey (frozen and defrosted bloodworm) over time when dispersed and aggregated, and in clear (visual and olfactory cues available) and turbid (no visual cues available) water. Additionally, we tested the effect of three levels of aggregation in the field in order to include more naturally sized foraging settings and multiple predators.

Methods

Laboratory experiment—does turbidity affect best aggregation strategy?

Study species and housing

Three spined sticklebacks (36–46 mm total body length) were caught by netting from small water bodies in Saltfleet, Lincolnshire (53°25′59.55″N, 0°10′49.41″E) in November 2010 and 2011. On both occasions, 250 fish were caught and were transported in commercial fish bags to the aquarium facilities at the University of Leeds. Fish were housed in groups of approximately 50 in grey plastic tubs (60 × 90 × 45 cm) with gravel substrate and artificial plants for environmental enrichment, at 14 ± 2 °C and on a 14:10 h light:dark cycle. Fish were fed ad libitum on defrosted frozen bloodworm (chironomid larvae, these were also the prey species in the experiment) from a commercial fish food supplier once daily. Each group of fish was released one year after capture at the location where caught (in agreement with the Home Office and DEFRA).

Procedure

Our experimental procedure followed that in Johannesen, Dunn & Morrell (2012) and is briefly summarized here. We investigated two levels of prey aggregation (aggregated and dispersed) and two levels of water clarity (clear and turbid) in a crossed design, giving 4 treatments (clear-aggregated, clear-dispersed, turbid-aggregated and turbid-dispersed). For logistical reasons, treatments were carried out in a semi-systematic order of blocks of 2–4 experiments of one treatment, followed by a block of another treatment. In each trial, eight designated locations in a foraging arena (100 × 100 cm, depth 5 cm, with a 10 × 10 cm central floating polystyrene shelter) received either one (dead) prey each (dispersed prey) or eight prey in one location (aggregated prey) chosen at random. Each location was a distance of 25 cm from the nearest neighbours and 25 cm from the arena wall. Turbid water was created by the suspension of commercial clay (Low Temperature White clay from Commercial Clay Ltd) in conditioned water at 0.5 g/l. Clay is commonly used to create turbidity without ill effects in the study animals (Ferrari, Lysak & Chivers, 2010; Vollset & Bailey, 2011). Fish showed no signs of distress in turbid water (gill flares, increased respiration or decreased motivation to feed) and trials lasted no more than 35 min. Water was changed between trials to remove olfactory cues from previous fish or prey, and fish were starved for 24 h before testing to standardize motivation to feed. As our aim was to investigate how prey aggregation affects olfactory prey detection by predators and how survival is affected by prey group size, we chose to use immobile (dead) prey. Mobile prey could produce other cues (e.g., lateral line detection) and potentially benefit from other mechanisms than dilution of risk (e.g., confusion). Testing these other factors was not within the scope of our study.

Trials were video recorded from above. In each trial a single fish was released under the floating shelter to acclimatize and time to emerge (be fully free of the shelter) was recorded. Fish were only used once and those that did not hide under the shelter on release or did not emerge within 15 min were excluded from the experiment (final sample sizes; clear water and aggregated prey: N = 13, clear water and dispersed prey: N = 15, turbid water and aggregated prey: N = 13, turbid water and dispersed prey: N = 15). Turbidity in the arena decreased over time, from 391.15 ± 9.35 NTU before fish were released to 286.83 ± 9.1 NTU after 35 min (measured before fish were captured after the trial). To ensure that visibility remained low in turbid water trials, fish were given a maximum of 35 min in the foraging arena, consisting of up to 15 min before emergence, plus 20 min during which foraging was recorded. We assessed the effect of environment (clear/turbid) on time to emergence using a negative binomial GLM to account for overdispersion in the Poisson-distributed data. There was no effect of environment on time to emergence (z = −1.63, df = 61, P = 0.1). This suggests that our manipulation of visual cues did not influence motivation to hunt for prey and/or perceived predation risk of the fish.

Data on foraging behaviour and time of prey capture for each prey item were manually extracted from videos using Etholog (2.25) and Windows Media Player. Sticklebacks vary considerably in boldness (Ward et al., 2004; Frost et al., 2007; Harcourt et al., 2010), leading to variation in time spent hiding (and therefore not foraging). Thus, to standardize search time for all fish, we recorded prey capture as a function of time spent actively swimming.

Field experiment: do prey in a more natural setting benefit from aggregating?

Our laboratory experiment necessarily constrained the search area available for each predator, increasing the likelihood of chance encounter. Furthermore, it tested the effect of aggregation of prey on survival, but was limited by the small number of prey. As predators were able to consume all prey without reaching satiation, our experiment did not include factors such as the dilution of individual risk (Wrona & Dixon, 1991) once discovered. In ponds and lakes, search volume or area is much greater, and there may be multiple predators (individuals or species) in the environment, affecting how many prey may be consumed and increasing the likelihood of local or stimulus enhancement (where the activity of an individual draws the attention of an observer towards a location or object; (Spence, 1937; Thorpe, 1956)), or social learning (Brown & Laland, 2003). To test the real-world validity of some of our findings, we also carried out a field experiment to assess the survival of visually hidden prey at different levels of aggregation. In order to ensure that cue availability was high enough in these larger water bodies, more prey were used. Because of this, aggregated and semi-dispersed prey groups were large enough to satiate a single predator, thereby allowing for dilution of individual risk within the experiment. The difference in setting and prey number make these two studies complementary rather than directly comparable.

Fieldwork was carried out in fresh to brackish water rock pools on the Faroe Islands, where there is a low diversity of aquatic species, making natural systems much simpler than those in warmer climates (Malmquist et al., 2002; Brodersen et al., 2011). The largest predators in a typical pool above the tidal line are Gammarus duebeni (Roberts, 1995) and sometimes three spined sticklebacks (Gasterosteus aculeatus). These ponds also contain a range of invertebrate prey species, including midge larvae which could be found in all ponds included in this experiment. Ponds (N = 11) were 5–50 m2 in size, all contained sticklebacks, some contained Gammarus, and none connected directly to any other pond in the study. Turbidity in these ponds varies naturally, but was low during our trials (below 10 NTU for all ponds). Visual cues were blocked with the use of “feeding stations” with opaque walls that allowed for transmission of olfactory cues.

Procedure

We created “feeding stations” to conceal visual, but not olfactory, cues from prey. Each feeding station consisted of a weighted transparent cylindrical plastic frame (12 cm diameter, 8 cm height) covered in two layers of fine-mesh material (nylon tights, 40 denier) with two entrance holes (2 × 2 cm) positioned at opposite sides of the station (Fig. 1). The stations were constructed in this way to allow olfactory cues to pass through the sides of the stations freely (pilot experiments in the lab with food dye indicated that cues passed through the walls). Cue movement is extremely slow in still water (Webster & Weissburg, 2009), but movement of fish and the disturbance caused by the experimenter moving the station to count prey enhanced cue dispersal. Disturbance was equal across all feeding stations (see below). In each pond, we placed 6 stations close to the edge (10–30 cm, to allow access by the experimenter), approximately 1 m apart. Stations were added 2–4 days prior to the first observation day to counter any effects of neophilia or neophobia (Frost et al., 2007; Archard & Braithwaite, 2011). To reduce disturbance, feeding stations were left in the ponds for the duration of the trials.

Figure 1 Feeding station used in the field.

“Feeding station” after use in field trials. Cotton thread attached at the top assisted in positioning and retrieval of stations and to the right is an entrance hole with “doors” intact to ensure opening was not blocked by stray material. A similar opening is found on the opposite side of the station.

In each pond, we investigated three levels of prey aggregation: (1) aggregated where 30 prey were placed in one of the 6 feeding stations while the remaining 5 stations remained empty, (2) semi-dispersed with 10 prey in each of 3 of the 6 stations and 3 empty stations, and (3) dispersed prey where we placed 5 prey in each of the 6 stations. Aggregated prey were allocated to a feeding station at random and semi-dispersed prey were allocated to alternating feeding stations (starting point chosen at random). The order in which the treatments were placed in each pond was systematically rotated ensuring each possible trial sequence was included at least once and no more than twice. To minimize any possible effects of learning and to reduce disturbance, a minimum of 4 days was left between each trial within a pond. Prey used in these trials were frozen bloodworm sourced from a local pet shop. The bloodworm were defrosted and the refrozen in tap water ice cubes in the prey groups sizes above for ease of handling in the field.

On the day of each trial, the ice cubes containing prey were positioned in their allocated feeding stations. Plain ice cubes (containing no prey) were placed in all other stations to control for the presence of the observer at each station and any cues from the tap water that may have been used by potential predators. After 10, 20, 30, 40, 50, 70 and 90 min, the observer returned to the pool and counted the number of uneaten prey in each station. Stations containing no prey were also checked to control for the presence of the observer and the disturbance caused by removing and replacing the feeding station. The timer was stopped when the observer returned to the pool, and restarted when counting was complete (approximately 10 min), so that the time while disturbed by researcher was not included in the time available to the fish to forage in the stations. It is likely that the presence of the observer disrupted normal foraging behaviour, so care was taken to ensure that this disruption was equal for all treatment groups. However, it is likely that detection would be faster than our data suggests due to this disruption. For this reason, we do not presume to make any claims about absolute detection times, but rather relative differences between prey group sizes in this study.

Analysis

All data analysis was carried out in R v 2.13.0 (R Core Team, 2013). For the laboratory data, prey within a trial were not independent of one another. To account for this, we created multiple events (each predator could consume multiple prey ‘events’) models using the Andersen-Gill version of Cox Proportional Hazards models in the package ‘survival’ (Therneau & Grambsch, 2000; Therneau & Lumley, 2011). Using this method, each prey item is considered an observation (whether consumed or not) leading to model sample sizes referring to individual prey items (8 per trial) rather than the predator or trial numbers. By clustering on ‘trial’ (this is akin to adding a random effect), we include in the model that individual prey within trials are not independent.

Our initial analysis of the laboratory data using a Cox Proportional Hazards model did not meet the necessary assumption of proportional hazards (Chi-squared = 85.6, P < 0.001; (Therneau & Grambsch, 2000)). When this assumption is violated, it is an indication that the survival curves are not the same shape and do not follow similar hazards distributions (i.e., the risk to a prey individual in one treatment is not a simple multiplication of the risk in another treatment, for any given time point). This is especially problematic when survival curves cross as they do in our case; Fig. 2 (Therneau & Grambsch, 2000). In order to remedy this, we split our data set in two (“initial prey discovery” and “subsequent survival of prey”) and analyzed these separately (Fig. 3). The assumption of proportional hazards was met in the case of initial prey discovery (Chi-squared = 3.27, P = 0.351). In the case of subsequent prey discovery, the assumption of proportional hazards was not met (Chi-squared = 176.4, P < 0.001). However, survival curves did not cross (Fig. 3B), so although predictions based on this model should be treated with caution (Therneau & Grambsch, 2000), it does give an indication of whether the survival of prey differed between treatments. Here, we use the term ‘survival’ in the context of survival analysis, where a prey individual ‘survives’ if it avoids being consumed by a predator (in reality, all prey in our experiments are already dead).

Figure 2 Overall prey survival in the laboratory.

Kaplan–Meier survival curves for the four groups of prey. Crosses signify censored events where the observations for a particular trial ended before all prey were eaten. The curve for aggregated prey in turbid water shows a different pattern to the curves for the other three treatments.

The data from field trials were interval censored, meaning the exact time of each prey being eaten was not known. Times were defined as the start and stop time of the interval in which prey were eaten, and we fitted a non-parametric maximum likelihood estimate (NPMLE) of the survival distribution (Turnbull, 1976). Hypothesis testing was performed using a non-parametric logrank test, using the packages ‘interval’ and ‘icens’ developed for analyzing interval censored data (Fay & Shaw, 2010; Gentleman & Vandal, 2011). Sun’s scores (Sun, 1996) indicate the differences and direction of difference between groups, and are chosen for analysis as they are flexible with respect to the duration between interval censored events and are the default option in the ‘interval’ package for this reason (Fay & Shaw, 2010).

Ethical statement

As experiments with fish fall outside of the remit of the University of Leeds Ethical Board and no licensed procedures were used, this study was not subject to ethical review.

However, laboratory experiments were carried out in accordance with University of Leeds guidelines and in agreement with Home Office licensed technical staff at the animal facility. Similarly, field experiments were carried out in accordance with local laws and regulations. Great care was taken to ensure optimal welfare for all fish involved in this study.

Results

Laboratory experiment—does turbidity affect best aggregation strategy?

The survival curve for aggregated prey in turbid water showed a very different pattern to the survival curve for other treatment groups (Fig. 2). The assumption proportional hazards was not met (Chi-squared = 85.6, P < 0.001; see above). This suggests that overall patterns of survival differ significantly as a function of treatment grouping.

When detection of first and subsequent prey are analyzed separately, it is clear that aggregation is beneficial in increasing the time to initial detection in both clear and turbid water, but has a greater effect in turbid water. There was a significant interaction between water clarity and level of aggregation (CoxPH; z = 2.24, n = 56, P = 0.025) on the time until the first prey was discovered (Fig. 3A). Dispersed prey are discovered more quickly in turbid water than clear water while aggregated prey are discovered more quickly in clear water than turbid water (Fig. 3A).

Figure 3 Survival on first prey in trial (A) and subsequent prey (B).

Kaplan–Meier curves for time to discovery of first (A) and subsequent (B) prey. Brown lines represent turbid water and blue lines clear water. Solid lines represent aggregated prey and dashes represent dispersed prey. In (B), the time axis was logged to improve clarity.

Figure 4 Field trial survival.

Interval censored survival curves for the field data. Possible stepwise changes in survival lie within the shaded area for each curve. Aggregated: solid line, light shading, semi-dispersed: dashed line, medium shading, dispersed: dotted line, dark shading.

For time to consume subsequent prey, there was also a significant interaction between the water clarity and level of aggregation (CoxPH, z = −3.173, n = 302, P = 0.002). Survival is highest for dispersed prey in turbid water, while aggregated prey survive longer in clear water than in turbid water (Fig. 3B). Therefore, after the discovery of the first prey, aggregation appears to be beneficial in clear water (aggregated prey survive longer in clear water than in turbid water), but not in turbid water (where dispersed prey have higher survival).

Field experiment: do prey in a more natural setting benefit from aggregating?

In the field experiment, prey in the three levels of aggregation differed significantly in survival (nonparametric log-rank test with Sun’s scores, Chi-squared = 13.16, P = 0.001 (Fig. 4)) with dispersed prey being discovered and consumed the most quickly and little to no difference between aggregated and semi-dispersed prey (Suns’ score statistics: dispersed: 42.17, aggregated: −19.11, semi-dispersed: −23.06).

Discussion

The data gathered both in the laboratory and in the field reveal that aggregation as a predator avoidance strategy is effective both for visually conspicuous and concealed prey.

Aggregated prey in the lab, with and without visual cues available to the predator, had improved survival (i.e., were less likely to be consumed) over dispersed prey in terms of initial detection. However, once an aggregation was detected, the prey did not survive (avoid consumption) for very long. This likely occurred because predators were able to find and consume all the prey in an aggregation after having discovered the first prey, and the dead prey could not take any evasive action in response to the proximity of the predator.

In the natural pond setting, overall survival of aggregated and semi-dispersed prey was higher than that of dispersed prey. Additionally, the rapid decrease in aggregated prey numbers once discovered in the lab was not observed in the field. This lack of sudden mortality post discovery is likely due to the large number of prey satiating the predator and thereby providing dilution of risk.

Due to the necessary differences in design between our field and laboratory experiments (see methods), we discuss our results within experimental context rather than making direct comparisons between the field and lab data.

In the field, we observed that prey removal in non-aggregated treatments was dispersed between stations, indicating that fish were not clearing out one station and then swimming to the next. The overall poorer survival of dispersed prey compared to semi-dispersed and aggregated prey suggests that aggregation should be an adaptive strategy for species living in water where visual cues are limited or absent as well as where the predator of immediate concern does not use visual cues.

Aggregation as an anti-predator strategy when the predator does not use visual cues is seen in a number of species such as the sediment dwelling Chironomus riparius larvae, who aggregate in response to predator presence (Rasmussen & Downing, 1988) and stream dwelling caddis flies (Rhyacophila vao) that avoid predation by the planarian predator Polycelis coronata by communally pupating on the same stone (Wrona & Dixon, 1991). Taylor’s (1977) study on southern grasshopper mice found that buried aggregated prey were found less easily than dispersed prey. Our data indicate that aggregation can be beneficial to prey in decreasing risk of detection, but also that aggregation is only truly effective if aggregations are large enough to dilute predation risk once discovered if prey are immobile. In many systems prey are at risk from several species of predators and aggregation in response to one predator may be counter-productive if another predator is present (Long et al., 2007). Showing behavioural flexibility in response to predator presence may be a better strategy than simply aggregating by default (Reimer & Tedengren, 1997; Kobak, Kakareko & Poznańska, 2010).

There is evidence in our lab results to suggest that the protection provided by aggregating depends partly on the availability of visual cues as well as the perception of risk in the predator. Once discovered, aggregated prey did not survive for long, but those in clear water survived for longer than those in turbid water. Although time to emergence was not affected by turbidity, we suggest that a perceived risk involved in foraging in clear open water (Abrahams & Kattenfeld, 1997) and the decreased vigilance resulting from foraging activities (Brown, 1999) combined to reduce foraging effort and allowed aggregated prey to survive longer once discovered in clear water than in turbid water.

In the field, aggregated prey did not experience the accelerated death rate once discovered that they did in the laboratory. There is some indication that benefits to prey depend on size or number of predators (Brock & Riffenburgh, 1960) and sticklebacks are able to learn from visual foraging cues from conspecifics (Webster & Laland, 2012), resulting in increased discovery if one stickleback in the group starts consuming prey. However darkness or turbid water should reduce the likelihood of this happening, as initial discovery of prey by one predator would not be observed visually by other predators. Lateral line detection of the movement of conspecifics (Coombs, 1999) is likely to be too short-range to be relevant in this context, however the importance of noises generated by foraging might warrant further exploration. In our experiment, prey as well as any predator feeding on them, were concealed in feeding stations, which may have prevented visual social cues from being transmitted to other sticklebacks in the area. Prey groups were also much larger than in the laboratory, which likely prevented individual sticklebacks from consuming all prey. Together, this may have limited the rapid consumption of prey seen in the laboratory.

The benefits of aggregation are likely to depend on the sensory abilities of the predator and a predator that is unable to detect prey will approach random search efficiency (Cain, 1985). However, a predator that is able to detect the presence of prey and perhaps even an indication of the number of prey should perform better than random by increased search effort, especially if that effort can be focused in the general area surrounding prey. Sticklebacks use both visual and olfactory cues in foraging, and when visual cues are not available, the presence of olfactory cues increases foraging efficiency (Johannesen, Dunn & Morrell, 2012). Therefore, strong cue concentrations around aggregated prey could increase search effort, potentially countering the benefit prey derive from aggregating. Similarly, theory on the relationship between olfactory cues and detection of prey groups predicts that grouping should not be favoured as detection radius increases with group size (Treisman, 1975). In our study, however, it is clear that aggregation is beneficial to prey, at least at the predator–prey ratios tested here, as our aggregated prey survived for longer than the dispersed prey. There is some evidence to suggest that olfactory detection radius increases with group size (Andersson, Löfstedt & Hambäck, 2013), but it is still not clear how increased detection affects aggregated prey in different systems such as one where only one prey item is captured and the rest escape and how predator sensory acuity interacts with prey group sizes.

Aggregations are ubiquitous and part of many important life functions. Understanding detectability and survival of aggregated prey will help us understand the adaptive mechanisms driving distributions of prey organisms and how these interact with predators. Our study provides insight into some adaptive reasons to aggregate in a system that is different from the usual visual predator system. Many natural predators rely on olfactory cues but the consequences of this have been relatively neglected by scientists, likely because of the dominant importance of vision to humans. We demonstrate that aggregations are beneficial to prey avoiding non-specialist olfactory foragers. Since predation is a fundamental interaction structuring communities, changes in the relative importance of vision and olfaction in prey detection (due to e.g., eutrophication) could have far reaching implications ecologically. Our work provides a step towards improved ability to predict these effects.

We wish to acknowledge Graeme Ruxton for valuable feedback on this manuscript. We also wish to acknowledge Charlotte Leviston and Hugin Kárason Mortensen for their invaluable help in gathering data for the laboratory and field studies respectively. Finally, we wish to acknowledge Delbert Smee and Rachel Lasley-Rasher for helpful comments on our manuscript.

Additional Information and Declarations

Competing Interests

Author Contributions

Animal Ethics

Data Deposition

The authors declare there are no competing interests.

Asa Johannesen conceived and designed the experiments, performed the experiments, analyzed the data, contributed reagents/materials/analysis tools, wrote the paper, prepared figures and/or tables.

Alison M. Dunn reviewed drafts of the paper, provided feedback and helpful comments before and during field work.

Lesley J. Morrell contributed reagents/materials/analysis tools, reviewed drafts of the paper, provided feedback on methodology and advice during data collection.

The following information was supplied relating to ethical approvals (i.e., approving body and any reference numbers):

As experiments with fish fall outside of the remit of the University of Leeds Ethical Board and no licensed procedures were used, this study was not subject to ethical review.

However, laboratory experiments were carried out in accordance with University of Leeds guidelines and in agreement with Home Office licensed technical staff at the animal facility. Similarly, field experiments were carried out in accordance with local laws and regulations. Great care was taken to ensure optimal welfare for all fish involved in this study.

The following information was supplied regarding the deposition of related data:

The full dataset is on FigShare: http://figshare.com/articles/Dataset_for_aggregation_study/801808.

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
