# Peer review of "Prey aggregation is an effective olfactory predator avoidance strategy"

_PeerJ, doi:10.7717/peerj.408_

## Round 0.1 · original submission · Minor Revisions

Both reviewers provided careful, detailed reviews, and both recommended minor revision before a final decision is made regarding acceptance. They each provide detailed and useful suggestions. Please revise in light of these reviews and provide a detailed response of how you addressed each comment. I suggest that you simply copy and past the reviews and below each comment insert in a different colored text (or in italic, etc.) how you addressed each comment/suggestion. You need not agree with every comment, but if you don't agree and did not alter your text, you do need to explain why. If the manuscript is carefully revised to address reviewer concerns, I expect to be able to make an editorial decision without additional review.

Cheers,
mark

·

Basic reporting

Johannesen et al. have performed a study to examine how prey aggregation influences predation rates in both clear and turbid water. Their findings indicate that aggregation can be beneficial to prey under certain conditions, namely water clarity. When the water is turbid, aggregation seems to benefit prey by delaying the time to initial detection, but, after detection leads to a more rapid consumption rate. This is likely caused by using deceased prey that obviously cannot flee when found. In the field, aggregation seemed to deter predation overall.
In summary, I found the study to be scientifically sound and encourage its acceptance for publication. The design and analysis are appropriate, and I liked the combination of lab and field experiments that complemented each other. There has been little work on the effects of turbidity on predatory interactions, and none I am aware of that combine turbidity and prey aggregation effects on predation. Below I offer a few suggestions to improve the manuscript for the authors and editors to consider. Overall I liked the study and found the paper easy and enjoyable to read. It was certainly interesting enough to keep me reading until the end.
Edits/Suggestions
Please add a comma after diversity in the first line of the abstract.
The third sentence in the abstract, beginning with ‘While the costs…’ is poorly worded. Try “Understanding how prey aggregation influences predation rates when visual cues and restricted, such as in turbid water, has not been thoroughly investigated.” Or something similar
The fifth sentence beginning with “In the laboratory…” is run on sentence that 5 lines long. Try breaking it into 2-3 sentences. I suggest adding a period after ‘available.’ Then, “Aggregated prey … in the longer term.” Then the statement in parentheses can be stated without parentheses to read “We attribute this finding to using dead prey that were thus unable to take evasive action to avoid predators.” Or similar
Line 30 Theory predicts? What theory? Please reword and just state plainly the theoretical relationship between group size and apparency to visual predators.
Lines 33-37 and worded clumsy. I don’t see how birds hunting for distasteful prey is like people hunting for computer generated prey. I also think this statement could easily be 2-3 sentences.
Lines 56-57 Please consider changing this to be 2 sentences.
In the methods, please add a statement to clarify replication. How many were done and was each fish used once or multiple times? In other words, is each replicate independent? I assume yes, if no, the analysis may not be correct.
I also was wondering why you measured fish size but never reported it? Was there a relationship between size and consumption or aggregation effects? Or, were you trying to standardize the size of the predators? If the latter, please state what the size range was clearly on line 113.
Please delete the word total from line 126
Line 143 I was confused about the ‘tidal line.’ I thought your pond was freshwater. Please make this more clear in the methods. If this is freshwater (I thought midges were only freshwater), the term tidal line is somewhat confusing.
Line 238 Please delete the comma after the parenthesis and add a period, then start a new sentence.
Line 248 Please remove the semi colon and add a period to create 2 sentences.
In the discussion, consider putting your study in a broader context by adding some text and references discussing how aggregation might be helpful for some predators but not others. You sort of get at this on lines 305-313. I would recommend expanding that section a bit to make the work of broader relevance. Jeremy Long has a nice paper in PNAS that deals with conditions under which aggregation is beneficial or not. (Long et al. 2007, Chemical cues induce consumer-specific defenses
in a bloom-forming marine phytoplankton.) This paper will contain some more references to add to better expand the discussion.

Experimental design

Appropriate design and analysis

Validity of the findings

Findings are interesting and unique as stated above

Additional comments

I found the study well done and the paper enjoyable to read. Nice work.

·

Basic reporting

Overall, the paper is clearly written. The authors provide appropriate background information on the benefits of prey aggregation and clearly outline how their study fits into this larger body of work. The paper is well structured with informative and appropriate figures. I have made a few minor comments regarding areas that require clarification. See ‘general comments for the author’.

Experimental design

There are some missing details regarding experimental execution and statistical analyses. For both laboratory and field experiments it is unclear how many replicate trials were conducted. Also, a clear description and justification of the statistical analyses used is needed. Please see 'general comments for the author' for more detailed remarks.

Validity of the findings

I was not familiar with all of the statistical tests used in this paper. After researching a bit, I agree with the authors choices regarding their statistical tests and therefore, accept their results as valid.

Additional comments

Overall, this paper is well written and the findings would be interesting to a broad audience. I have included minor comments below.

Line 63 – A justification for this prey species choice is needed here.
Line 91 – I assume that the trials were conducted one at a time. How many replicates trials were conducted? Did the authors randomize the order of treatments you administered? Were experiments conducted around the same time of day? Were fish re-used?
Line 92 – Change allocated to ‘received’
Line 96 – One could argue that if the clay somehow harmed the fish (i.e. disrupted respiration) that the fish feeding behavior would not be the same between your treatments. I do not think this is the case because I know this type of clay is often used in turbidity experiments. However, this may not be evident to all readers. Please state why you are confident that the clay did not have any harmful effect on the fish (i.e. your own observations or previous studies).
Line 113 – Did you use a negative binomial GLM due to overdispersion using a Poisson distribution?
Line 153 – Im not sure what is meant by “skeleton”.
Line 154 – Please report the mesh size of the nylon in micrometers.
Line 159 – You state that movement by the experimenter affected the olfactory cue transmission. How did you attempt to standardize this across treatments?
Line 171 -174– This information would be more clear if presented in a table.
Line 177 – Subject-verb confusion. Consider inserting ‘to’ reduce disturbance.
Line 192 – ‘not included in the final data’ is redundant with previous statement. Please remove.
Line 199-203 This description is unclear. How did you quantify prey encounters? The number of times a fish visited a station or conducted a prey strike, moved within a certain distance of the prey? Please clarify.
Line 205 – State what the assumptions are and which were violated.
Line 246 – Remove ‘data of’
Line 249 – Not sure what n = 56 refers to here. Is this the number of replicate trials or the number of individuals eaten? It seems that the level of replication should refer to the number of trials. The most conservative approach would be to compare the total number of prey eaten per trial across the 4 treatments. This approach avoids pseudoreplication and adheres more strictly to the concept an experimental replicate. Please clearly state how many times these trials were repeated and justify the level of replication you have chosen.
Line 260 – Remove ‘for’ from the second part of the sentence
Line 267 - Asymptotic Logrank k-sample test with Sun’s scores. Please explain this test and test statistics in the methods.
Line 269 – Please explain the Sun’s score in the methods section and give a citation.
Line 298 – Change reduction to removal or consumption.
Line 317 – The prey are already dead in all experiments. You should state in the methods that you use the word ‘survive’ to mean that a prey remains at the end of the trial to avoid confusion.
Line 319 – When an organism leaves a refuge to go forage they expose themselves to predation risk. When they engage in actual foraging aren’t they increasing this risk due to diverting their attention away from predator detection? You could cite examples of this to further bolster your argument here.

---

## Round 0.2 · accepted · Accept

You have done a good job of addressing the reviewer's concerns. Congratulations. The only issue I detected is that the indicator key for your treatments in Figure 3 is blocked by part of the figure. I assume that the Peer J staff will work with you to resolve this.